# Protein subcellular relocalization and function of duplicated flagellar calcium binding protein genes in honey bee trypanosomatid parasite

**Xuye Yuan, Tatsuhiko Kadowaki** [ORCID] *

Department of Biological Sciences, School of Science, Xi'an Jiaotong-Liverpool University, Suzhou Dushu Lake Higher Education Town, Jiangsu Province, China

* Tatsuhiko.Kadowaki@xjtlu.edu.cn

**Data Availability Statement:** All data are available in the submitted Supplementary data files.

**Funding:** This work was supported by Jinji Lake Double Hundred Talents Programme to TK. The

## Abstract

The honey bee trypanosomatid parasite, *Lotmaria passim*, contains two genes that encode the flagellar calcium binding protein (FCaBP) through tandem duplication in its genome. FCaBPs localize in the flagellum and entire body membrane of *L. passim* through specific N-terminal sorting sequences. This finding suggests that this is an example of protein subcellular relocalization resulting from gene duplication, altering the intracellular localization of FCaBP. However, this phenomenon may not have occurred in *Leishmania*, as one or both of the duplicated genes have become pseudogenes. Multiple copies of the *FCaBP* gene are present in several *Trypanosoma* species and *Leptomonas pyrrhocoris*, indicating rapid evolution of this gene in trypanosomatid parasites. The N-terminal flagellar sorting sequence of *L. passim* FCaBP1 is in close proximity to the BBSome complex, while that of *Trypanosoma brucei* FCaBP does not direct GFP to the flagellum in *L. passim*. Deletion of the two *FCaBP* genes in *L. passim* affected growth and impaired flagellar morphogenesis and motility, but it did not impact host infection. Therefore, *FCaBP* represents a duplicated gene with a rapid evolutionary history that is essential for flagellar structure and function in a trypanosomatid parasite.

## Author summary

Protein subcellular relocalization (PSR) was proposed as a mechanism for the functional divergence and retention of duplicate genes. We explored this hypothesis using flagellar calcium binding protein (FCaBP) genes duplicated in many trypanosomatid parasites. FCaBPs localize in the flagellum and entire body membrane of honey bee trypanosomatid parasite, *Lotmaria passim* through specific N-terminal sorting sequences. However, this is not common with all trypanosomatid parasites since one or both of the duplicated genes have become pseudogenes in *Leishmania*. N-terminal flagellar sorting sequence of FCaBP1 interacts with BBSome complex, and thus they must have co-evolved in each trypanosomatid species. FCaBPs are essential for the normal growth, flagellar morphogenesis, and motility but not host infection of *L. passim*. Our findings demonstrate that duplicated *FCaBP* genes have undergone PSR by changing amino acids in the N-terminal

funder had no role in study design, data collection and analysis, decision to publish, or preparation of the manuscript.

**Competing interests:** The authors have declared that no competing interests exist.

end with some but not all trypanosomatid parasites. In the parasites with multiple *FCaBP* genes, they play critical roles for the growth control as well as flagellar structure and function.

## Introduction

Gene duplication is considered the main source of new genes [1–4]. However, to maintain duplicate genes in a genome, functional divergence is usually necessary, with a few exceptions. Functional divergence can occur through neofunctionalization, where one of the duplicates develops a new function while the other retains the original function of the ancestral gene [1]. Alternatively, subfunctionalization can take place, where the ancestral functions are divided between the duplicate genes. For example, their combined levels or patterns of activity could be equivalent to the original single gene [5–7]. These types of functional divergence can modify the function of the encoded protein or the gene's expression pattern. Subcellular localization is also crucial for a protein's function within a cell. Protein subcellular relocalization (PSR) has been proposed as a mechanism for the functional divergence and retention of duplicate genes [8,9]. PSR allows for the rapid subdivision of ancestral localization or acquisition of new localization following gene duplication. After neolocalization or sublocalization, the expression pattern and function of the genes can further diverge. Although this idea was questioned by comparing the frequency of PSR between singletons and duplicates [10], there are numerous examples of PSR for duplicated genes [11–13].

The flagellar calcium binding protein (FCaBP or Calflagin) was discovered as one of the abundant proteins in the flagellar membrane of *Trypanosoma brucei* and *Trypanosoma cruzi*. It contains four EF-hand calcium binding domains and is considered a conserved signaling protein among trypanosomatid parasites [14,15]. FCaBP is acylated with myristate and palmitate at the N-terminus, which is crucial for its localization in the flagellum and association with lipid raft microdomains [16–18]. Additionally, the N-terminal amino acid sequence of FCaBP is necessary for its targeting to the flagellum in *T. cruzi* [19]. Although the phenotypes of knocking down FCaBPs in *T. brucei* have been reported [20], the physiological functions of FCaBPs are not well understood. Proteins in the flagellum (and cilium) are initially synthesized in the cell body and then transported to the flagellum through intraflagellar transport (IFT). IFT requires IFT complexes (IFT-A and IFT-B) and motor proteins, including kinesin-2 for anterograde transport and IFT-dynein for retrograde transport, along the axonemal microtubules [21–23]. The IFT complexes act as carriers, and BBSome and tubby-like protein (TULP) function as adapters by binding to specific sets of flagellar membrane proteins [24–29]. For example, BBSome is necessary for the exit of a dually acylated phospholipase D from cilia in *Chlamydomonas* [26]. Therefore, FCaBP likely requires IFT complexes and either BBSome or TULP for its import and export to the flagellum.

*Lotmaria passim* is the most prevalent trypanosomatid parasite that infects honey bees worldwide [30–33]. It specifically colonizes the hindgut of honey bees, affecting the host's physiology, and it may be associated with winter colony loss [34,35]. *L. passim* is a monoxenous parasite that only infects bees and can serve as a model to study the roles of flagellar formation and function for infecting the various hosts of trypanosomatid parasites.

In this study, we identified two *FCaBP* genes in the genomes of *L. passim* and closely related species, resulting from tandem duplication. By analyzing the gene in other trypanosomatid species' genomes, the protein subcellular localizations, and N-terminal sorting sequences, we have uncovered their evolutionary history in relation to the flagellar protein transport

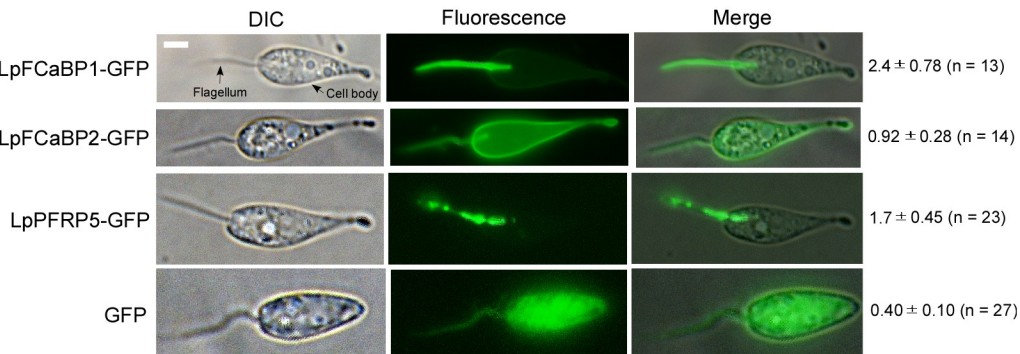

**Fig 1. Subcellular localization of LpFCaBP1- and 2-GFP fusion proteins.** *Lotmaria passim* expressing LpFCaBP1-GFP, LpFCaBP2-GFP, LpPFRP5-GFP, or GFP were examined under visible (DIC) or fluorescence (Fluorescence) light. Merged images are also shown. Flagellum at the anterior end and cell body are indicated by arrows. The ratio of the mean fluorescence in the flagellum to that in the cell body (mean value ± SD) along with number of parasites analyzed (n) is presented on the right. Scale bar: 2 μm.

machinery. Deletion of these genes in *L. passim* has also revealed novel physiological functions of FCaBPs. Therefore, *FCaBP* serves as an example to understand how duplicated genes have diversified during the evolution of trypanosomatid parasites.

## Results

### *L. passim* contains two FCaBPs with flagellar and entire body localizations

The genome of *L. passim* encodes two *FCaBP* genes (*LpFCaBP1* and *LpFCaBP2*) that are separated by 1063 bp, suggesting tandem gene duplication. To determine their localizations, we expressed GFP-tagged versions of LpFCaBP1 and LpFCaBP2 (at C-terminus) in *L. passim*. LpFCaBP1-GFP was highly enriched in the flagellum, similar to FCaBPs in *T. cruzi* and *T. brucei* [16,17]. On the other hand, LpFCaBP2-GFP was present at the flagellum and the cell body membrane (Fig 1). We also observed that GFP alone is uniformly distributed in the entire body, while the paraflagellar rod protein 5 (LpPFRP5)-GFP fusion protein is specific to the flagellum (Fig 1). These findings indicate that the two duplicated *FCaBP* genes encode proteins enriched in the different part of parasite body with common localization in the flagellum.

### The N-terminal 16 amino acids of LpFCaBP function as either flagellum or cell body membrane sorting sequence

By aligning the amino acid sequences of LpFCaBP1 and LpFCaBP2, we found that they are almost identical except for their N-terminal ends (S1 Data). To determine the flagellar and entire body membrane sorting sequences, we swapped the N-terminal 16 or 28 amino acids between LpFCaBP1 and LpFCaBP2 in GFP fusion proteins. We also determined the ratio of mean fluorescence of flagellum to that of cell body using Image-J. There are no statistical differences between the ratios for LpFCaBP1-GFP (2.4 in average, Fig 1), LpFCaBP1N28/ FCaBP2-GFP (2.3 in average, Fig 2), LpFCaBP1N16/FCaBP2-GFP (2.3 in average, Fig 2), and LpFCaBP1N16-GFP (2.1 in average, Fig 2). The ratio for LpFCaBP2-GFP (0.92 in average, Fig 1) is significantly higher than those of LpFCaBP2N28/FCaBP1-GFP (0.66 in average, $P < 0.003$, Fig 2), LpFCaBP2N16/FCaBP1-GFP (0.78 in average, $P < 0.04$, Fig 2), and LpFCaBP2N16-GFP (0.49 in average, $P < 0.001$, Fig 2). These results demonstrate that the N-terminal 16 amino acids of LpFCaBP1 and LpFCaBP2 are sufficient to direct GFP to the

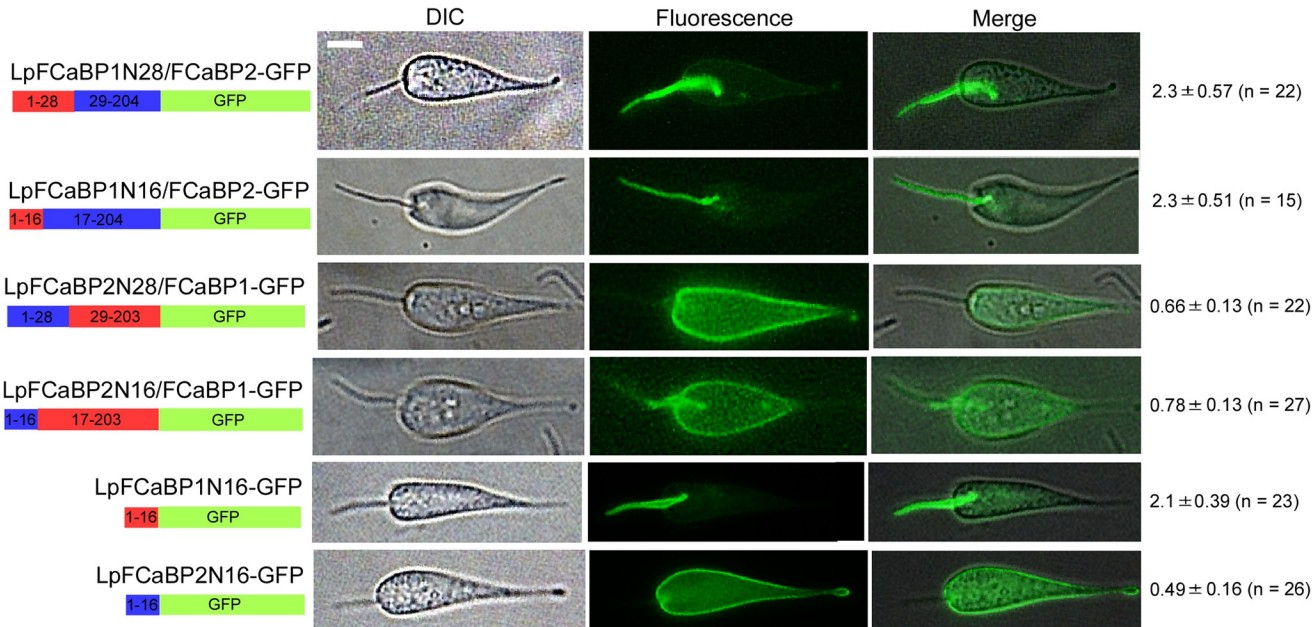

**Fig 2. Flagellum and cell body membrane sorting sequences of LpFCaBP1 and 2.** The N-terminal 16 or 28 amino acids of LpFCaBP1 (in red) and LpFCaBP2 (in blue) were swapped as illustrated in the diagram. The positions of amino acids in the cores of LpFCaBP1 and 2 proteins are also indicated. GFPs fused with the N-terminal 16 amino acids of LpFCaBP1 and 2 are shown at the bottom. The ratio of the mean fluorescence in the flagellum to that in the cell body (mean value ± SD) along with number of parasites analyzed (n) is presented on the right. Scale bar: 2 μm.

flagellum and cell body membrane, respectively (Fig 2). Thus, they represent the flagellar and cell body membrane sorting sequences. Since LpFCaBP2-GFP was present in the entire body (Fig 1), the EF-hand calcium binding domains (amino acids 38–189) contribute to the flagellar localization.

## FCaBP genes have rapidly evolved in trypanosomatid parasites

We examined *FCaBP* genes in various trypanosomatid parasites' genomes. *T. brucei* has four *FCaBP* and the related genes as previously reported [20]. *T. cruzi* appears to contain the large number of *FCaBP* genes due to its repetitive genome [36]. *Trypanosoma theileri* genome has three *FCaBP* genes (S2 Data). These results suggest that *FCaBP* gene duplicated in the common ancestor of *Trypanosoma* and Leishmaniinae and has undergone further duplication in each *Trypanosoma* species.

In the genomes of various Leishmaniinae species, microsynteny around *FCaBP* genes is well conserved with the same orders and orientations of genes encoding three novel proteins, dynein light chain, and sucrose-phosphate synthase-like protein (Fig 3). In *Leishmania infantum*, *Leishmania braziliensis*, *Leishmania mexicana*, and *Leishmania donovani*, two *FCaBP* genes are present but *FCaBP2* appears to have become a pseudogene by generating a stop codon in 5' end of the ORF (open reading frame). If this gene is translated, the truncated protein lacking the N-terminal sorting sequence is synthesized and is likely to be present in the cytosol rather than the membrane, as it lacks the N-terminal glycine and cysteine residues necessary for myristoylation and palmitoylation. Although the protein should be able to bind calcium, its interactions with other membrane associated proteins as a signaling factor would be lost. Thus, above four *Leishmania* species may contain only *FCaBP1*. Intriguingly, two *FCaBP*

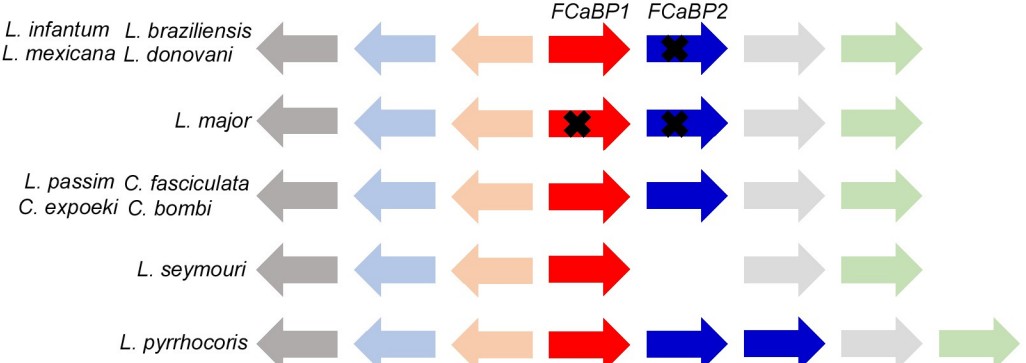

**Fig 3. Conserved microsynteny of *FCaBP* genes in Leishmaniinae species.** The microsynteny containing *FCaBP* genes is highly conserved in *Leishmania infantum, Leishmania braziliensis, Leishmania mexicana, Leishmania donovani, Leishmania major, Lotmaria passim, Crithidia fasciculata, Crithidia expoeki, Crithidia bombi, Leptomonas seymouri,* and *Leptomonas pyrrhocoris. FCaBP2* is a pseudogene (indicated by ×) in *L. infantum, L. braziliensis, L. mexicana,* and *L. donovani,* while both *FCaBP1* and *2* are pseudogenes in *L. major. FCaBP1* and *2* genes (red and blue arrows) are flanked by three novel genes (beige, light grey, and light green arrows), as well as dynein light chain (light blue arrow) and sucrose-phosphate synthase-like protein genes (dark grey arrow). The orientation of each gene is shown.

genes in *Leishmania major* have become pseudogenes by accumulating multiple internal stop codons in the ORFs (Fig 3). These characteristics are shared with three independently assembled genome sequences derived from different strains of *L. major* (Friedlin, SD75.1, and LV39c5), suggesting that this is not the result of genome assembly error.

In species related to *L. passim* (34), *Crithidia fasciculata, Crithidia expoeki,* and *Crithidia bombi* have two *FCaBP* genes, while *Leptomonas seymouri* has only *FCaBP1* gene (Fig 3). At downstream of *LsFCaBP1,* there is 173 bp genomic sequence capable of encoding a protein with low similarity to C-terminal 56 amino acid of FCaBP (e-value: 0.013) if translated. This may represent the remnant of *LsFCaBP2* pseudogene. We primarily characterized *L. seymouri* assembled genome sequence derived from the strain ATCC 30220 Lsey_0068 (Accession number: LJSK01000068); however, *FCaBP* genomic sequence is the same in another independently assembled genome sequence derived from the different strain, BHU1095 (Accession number: ANAF02000031). Thus, the loss of *LsFCaBP2* is unlikely to be caused by the error of genome assembly. *Leptomonas pyrrhocoris* contains *FCaBP1* and two *FCaBP2* genes based on the comparison of N-terminal sorting sequences (S1 Data). These results demonstrate that *FCaBP* gene has rapidly evolved in trypanosomatid parasites.

## Flagellar localization of FCaBP depends on both the sorting sequence and *trans*-acting factors in *L. passim*

Fig 4A shows the N-terminal ends of aligned FCaBPs from *T. cruzi, T. brucei, L. donovani,* C. *fasciculata, L. seymouri,* and *L. passim*. Glycine and cysteine residues targeted for myristoylation and palmitoylation are well conserved; however, there are more variations in the sorting sequences of FCaBPs compared to the main sequences with EF-hand calcium binding domains. To elucidate the pivotal amino acids within the sorting sequence governing the membrane localization of LpFCaBP in the flagellum or entire body, we performed targeted mutagenesis on LpFCaBP1 and LpFCaBP2. Specifically, lysine at position 11 (K11) in LpFCaBP1 was substituted with isoleucine, and vice versa, generating LpFCaBP1N16(K11I)-GFP and LpFCaBP2N16(I11K)-GFP. Additionally, serines at positions 9 (S9) and 10 (S10) in

**A**

| | | | | |
|---|---|---|---|---|
| TcFCaBP | MGACGSKGS----TSDKGLASDKDG | KNAKDRKEAWERIRQAIPREKTAEAKQRRIELFKK | 56 |
| TbFCaBP | MG-CSGSKNTTNSKDGAASKGGKDG | KTTADRKVAWERIRCAIPRDKDAESKSRRIELFKQ | 59 |
| LdFCaBP1 | MG-CNAT---------KAARKPSED | KTAADRKVAWEKICQRLPRQKTPEDKELHIELFKR | 50 |
| CfFCaBP1 | MG-CASSAF----SS----KSKKEG | KNASDRKAAWEGIRERLPRKKTDEDKERRIELFKK | 51 |
| CfFCaBP2 | MG-CISSKS----TQ----TGKKEG | KTAAERKAAWEGIRQRLPRRKTAEDKARRIELFKK | 51 |
| LsFCaBP1 | MG-CASSTS----SS---KGGKKEG | KSAAERRAVWGSVRQSLPRLKTAVDKERRIALFKE | 52 |
| LpFCaBP1 | MG-CASSLF----SS----KSKTED | KTAAERKVAWEKIRERLPRRKTPEDKERRIELFKK | 51 |
| LpFCaBP2 | MG-CISSKS----TQ----IGKKEC | KTAAERKAAWEGIRQRLPRRKTPEDKQRRIELFKK | 51 |
| | ** * .. | .: *.: :*: .* : :** * * :* ***. | |

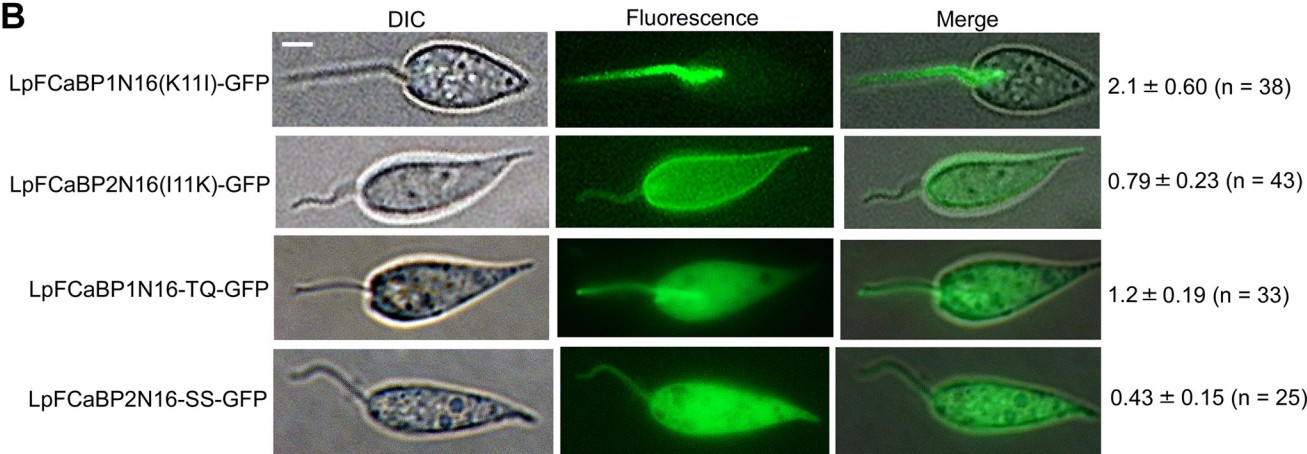

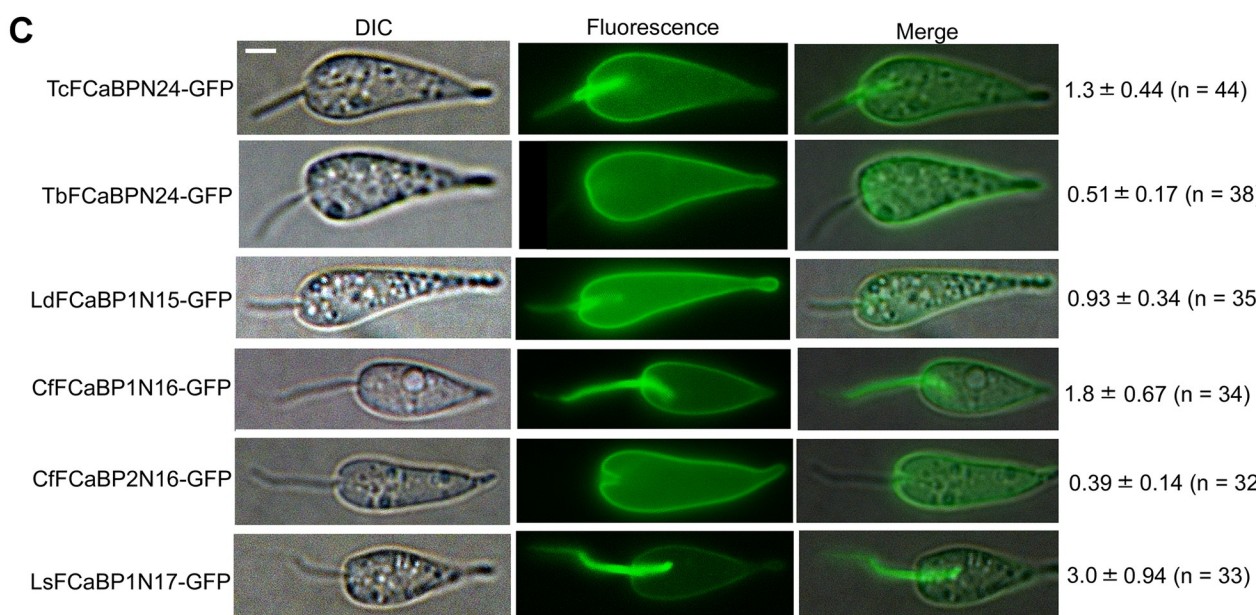

**Fig 4. Subcellular localization of GFPs with N-terminal sorting sequences of FCaBPs from various trypanosomatid parasites.** (A) N-terminal parts of the aligned sequences of *Trypanosoma cruzi* (AAA99985.1), *Trypanosoma brucei* (XP_847374.1), *Leishmania donovani* (XP_003859779.1), *Crithidia fasciculata*, *Leptomonas seymouri* (KPI87939.1), and *Lotmaria passim* FCaBPs are shown. The amino acids fused to GFP in (C) are indicated in bold letters. Glycine (red) and cysteine (blue) residues are targeted for myristoylation and palmitoylation, respectively. Conserved amino acids are shown by asterisks, and similar amino acids are indicated by either a full stop (.) or a colon (:). (B) Subcellular localization of GFPs fused with the mutated N-terminal sorting sequences of LpFCaBP1 and 2. (C) Subcellular localization of GFPs fused with the N-terminal sorting sequences in (A). The ratio of the mean fluorescence in the flagellum to that in the cell body (mean value ± SD) along with number of parasites analyzed (n) is presented on the right. Scale bar: 2 μm.

LpFCaBP1 were substituted with threonine and glutamine, respectively, and vice versa, resulting in LpFCaBP1N16-TQ-GFP and LpFCaBP2N16-SS-GFP. Comparative analyses with wild-type LpFCaBP1N16-GFP (Fig 2) revealed a substantial increase in cell body fluorescence and a corresponding decrease in relative flagellar fluorescence with LpFCaBP1N16-TQ-GFP (average 1.2, *P < 0.001*), while no significant change was observed with LpFCaBP1N16(K11I)-GFP (Fig 4B). In comparison to wild-type LpFCaBP2N16-GFP (Fig 2), LpFCaBP2N16(I11K)-GFP exhibited an increase in relative flagellar fluorescence (average 0.79, *P < 0.001*), while no significant difference was noted with LpFCaBP2N16-SS-GFP (Fig 4B). These results indicate that K11 of LpFCaBP1 is not essential to direct LpFCaBP1N16-GFP to the flagellum but contributes to the flagellar localization of LpFCaBP2N16-GFP. Furthermore, S9 and S10, which are conserved among FCaBP1s across four trypanosomatid species (S1 Data), are crucial for the efficient localization of LpFCaBP1N16-GFP to the flagellum in *L. passim*. Remarkably, the fluorescence pattern of LpFCaBP1N16-TQ-GFP and LpFCaBP2N16-SS-GFP is uniform within the cell body, akin to GFP (Fig 1), in contrast to the membrane localization observed with LpFCaBP2N16-GFP (Fig 2) and other GFP fusion proteins (Fig 4B and 4C). Thus, it is likely that S9, S10, T9, and Q10 are required for efficient acylation at glycine (G2) and cysteine (C3) in the N-termini of FCaBP1 and FCaBP2. These lipid modifications appear to play a crucial role in determining FCaBP localization, either in the flagellum as previously reported [17,19] or in the entire body membrane. However, LpFCaBP1N16-TQ-GFP and LpFCaBP2N16-SS-GFP may fail to associate with the cell body membrane due to a mechanism other than defective acylation.

To test the conservation and plasticity of the flagellar and cell body membrane sorting sequences of FCaBPs, we expressed GFP fusion proteins containing N-terminal sorting sequences from various trypanosomatid parasites in *L. passim*. Because TcFCaBP and TbFCaBP (Calflagin, Tb24) are flagellar proteins, their N-terminal amino acids should function as flagellar sorting sequences in *T. cruzi* and *T. brucei*, respectively [17,19]. Indeed, the N-terminal 24 but not 12 amino acids of the TcFCaBP were sufficient to direct GFP to the flagellum of *T. cruzi* [19]. However, we found that GFP fused with the sorting sequence of TbFCaBP is localized in the cell body membrane of *L. passim*. GFP with the sorting sequence of either TcFCaBP or LdFCaBP is present in both cell body membrane and the proximal part of flagellum (Fig 4C). The sorting sequences of CfFCaBP1 and 2 directed GFP to flagellum (most) as well as cell body membrane and cell body membrane, respectively. GFP with the sorting sequence of LsFCaBP1 is primarily localized in flagellum (Fig 4C). These results may suggest that flagellar localization of FCaBP requires the recognition of N-terminal sorting sequence by IFT-associated proteins (*trans*-acting factors).

## The N-terminal sorting sequence of LpFCaBP1 is in close proximity to the BBSome complex but not TULP

Since the import and export of flagellar membrane proteins depend on either the BBSome complex or TULP [37], we first tested the interaction between LpFCaBP1N16-GFP and Flag-tagged LpBBS1 or LpTULP in the parasites expressing both proteins by immunoprecipitation. However, we were unable to detect binding with either protein. This would be due to the transient interaction between a flagellar membrane protein and BBSome or TULP as the adaptor [37]. We thus used an engineered biotin ligase, ultraID [38], instead of GFP. This fusion protein allowed us to identify proteins in the close proximity through biotinylation. In *L. passim* expressing both LpFCaBP1N16-UltraID and Flag-tagged LpBBS1 or LpTULP, the ultraID fusion protein was detected in the proximal part of the flagellum (Fig 5A). Surprisingly, LpFCaBP2N16-UltraID concentrated at the anterior end of cell body (Fig 5A) in contrast to

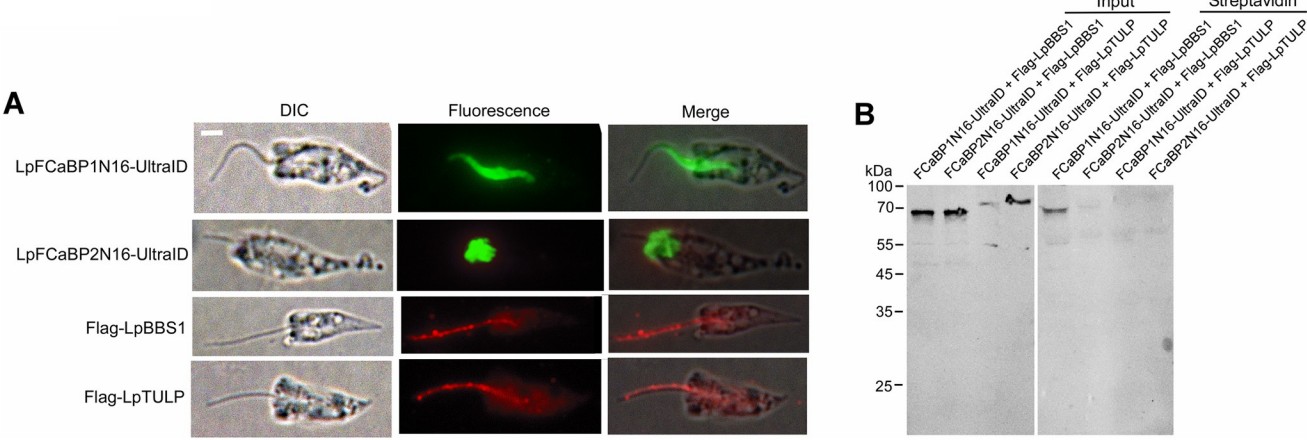

**Fig 5. Interaction of LpFCaBP1 N-terminal sorting sequence with LpBBS1.** (A) Subcellular localization of ultraID with the N-terminal sorting sequence of LpFCaBP1 (LpFCaBP1N16-UltraID), LpFCaBP2 (LpFCaBP2N16-UltraID), Flag-tagged LpBBS1 (Flag-LpBBS1), and LpTULP (Flag-LpTULP) by immunofluorescence. Scale bar: 2 μm (B) Lysates of parasites expressing both either LpFCaBP1N16-UltraID or LpFCaBP2N16-UltraID and Flag-LpBBS1 or Flag-LpTULP were subjected to immunoprecipitation using Flag Fab-Trap Agarose (Input) or streptavidin agarose precipitation (Streptavidin) to capture biotinylated proteins. One-fourth of the total protein in the Streptavidin sample was analyzed for the Input sample. The precipitates were analyzed by western blot using an anti-Flag tag antibody.

the uniform cell body membrane localization of LpFCaBP2N16-GFP (Fig 2). Thus, ultraID tends to anchor at the anterior side of cell body and this may also inhibit the localization of LpFCaBP1N16-UltraID to the distal part of flagellum in *L. passim* (Fig 5A). We detected biotinylated Flag-LpBBS1 but not Flag-LpTULP in the parasites expressing LpFCaBP1N16- but not LpFCaBP2N16-UltraID (Fig 5B), suggesting that LpFCaBP1N16 is in close proximity to LpBBS1. The apparent size of Flag-LpTULP was larger than that of Flag-LpBBS1 and the expected size (59 kDa) (Fig 5B). LpTULP may have post-translational modification or migrate slowly through 10% SDS-PAGE. These findings suggest that the flagellar localization of LpFCaBP1 may be mediated by the BBSome complex.

## LpFCaBPs are essential for flagellar morphogenesis and motility but not the host infection of *L. passim*

To determine the functions of LpFCaBPs, we deleted both *LpFCaBP1* and *LpFCaBP2* genes in *L. passim* by replacing them with hygromycin resistance gene by CRISPR-mediated homology-directed repair (Fig 6A). Although LpFCaBP1 and 2 primarily localize in the flagellum and entire body membrane, they could be still functionally redundant. As we previously reported, expressing Cas9 and gRNA does not modify the target gene in *L. passim*. Thus, the off-target effects are basically absent (39). By genomic PCR, we identified both heterozygous (+/-) and homozygous (-/-) deletion clones of *LpFCaBPs* (Fig 6B). To confirm the disruption of genes, we tested *LpFCaBP1* and *2* mRNA expression in wild type, heterozygous and homozygous deletion parasites by RT-PCR using the gene-specific reverse primer. As shown in Fig 6C, both mRNAs are absent in the homozygous mutant (*LpFCaBP1/2 -/-*) parasites.

We also established *LpFCaBP1/2 -/-* parasites, wherein either *LpFCaBP1* or *LpFCaBP2* was introduced on a plasmid DNA vector containing a bleomycin resistance gene. Comparative phenotypic analyses were performed with wild-type and the parent mutant parasites at 29˚C. *LpFCaBP1/2 -/-* parasites exhibited reduced growth rates during the logarithmic phase, reaching stationary phase at a density comparable to wild-type parasites. The growth defect in

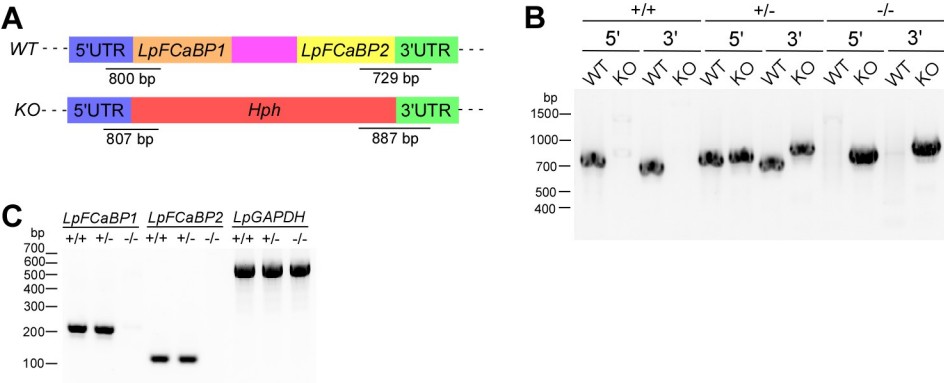

**Fig 6. Deletion of *LpFCaBP1* and *2* genes by CRISPR.** (A) Schematic representation of wild-type (*WT*) and deleted (*KO*) alleles of *LpFCaBPs* generated by CRISPR/Cas9-induced homology-directed repair. 5' and 3' untranslated regions (UTRs), open reading frames (ORFs), and the internal spacer of *FCaBP1* and *2*, hygromycin resistance gene *(Hph)*, are shown in blue, green, orange, yellow, purple, and red, respectively. The expected sizes of PCR products to detect *WT* and *KO* alleles (not to scale) are also shown. (B) Genomic DNAs of wild-type *L. passim* (+/+), heterozygous (+/-), and homozygous (-/-) mutants of *LpFCaBPs* were analyzed by PCR to detect *5'WT*, *5'KO*, *3'WT*, and *3'KO* alleles. Sizes of the DNA molecular weight markers are shown on the left. (C) Detection of *LpFCaBP1* and *2*, as well as *LpGAPDH* mRNAs, in *LpFCaBPs* heterozygous (+/-) and homozygous (-/-) mutants together with wild-type *L. passim* (+/+) by RT-PCR. Sizes of the DNA molecular weight markers are shown on the left.

*LpFCaBP1/2 -/-* parasites was rescued by the introduction of either LpFCaBP1 or LpFCaBP2 (Fig 7A and 7B). In the early stationary phase (at 60 h after culture initiation), wild-type parasites displayed active movement with extended flagella. Measurement of flagella and cell body lengths, as well as individual parasite movements, revealed that *LpFCaBP1/2 -/-* parasites exhibited shorter flagella and cell bodies with reduced motility compared to wild-type parasites (Fig 7C–7F and S1 and S2 Videos). These findings indicate the essential role of LpFCaBP1 and LpFCaBP2 in flagellar morphogenesis and motility of *L. passim*. While either LpFCaBP1 or LpFCaBP2 rescued the observed phenotypes, they did not fully complement to wild-type levels, except for LpFCaBP1, which fully restored the short flagella of *LpFCaBP1/2 -/-* parasites (Fig 7C–7F and S1–S4 Videos). In the late stationary phase (at 96 h after culture initiation), wild-type parasites formed rosettes, characterized by clusters of cells with flagella directed towards the center. However, *LpFCaBP1/2 -/-* parasites remained as individual cells without clustering at the culture plate bottom (Fig 7G). Both LpFCaBP1 and LpFCaBP2 restored rosette formation in *LpFCaBP1/2 -/-* parasites, although the sizes of rosettes were generally smaller than those observed with wild-type parasites (Fig 7G). These results suggest that the molecular functions of LpFCaBP1 and LpFCaBP2 remain the same as calcium-binding proteins. However, in *L. passim*, both LpFCaBP1 concentrated in the flagellum and LpFCaBP2 present throughout the entire parasite are necessary. We then compared the infection of wild-type and *LpFCaBP1/2 -/-* parasites in honey bee hindgut and found that they infect the host at the comparable level (Fig 7H). Thus, less motile *L. passim* with short flagellum by the loss-of function of *LpFCaBP* genes is still capable of infecting honey bee hindgut effectively.

## Discussion

### Rapid evolution of *FCaBP* genes in trypanosomatid parasites

We characterized *FCaBP* genes in the genomes of trypanosomatid parasites and observed that each species, except *L. seymouri*, has more than two *FCaBP* genes in the same genomic contigs

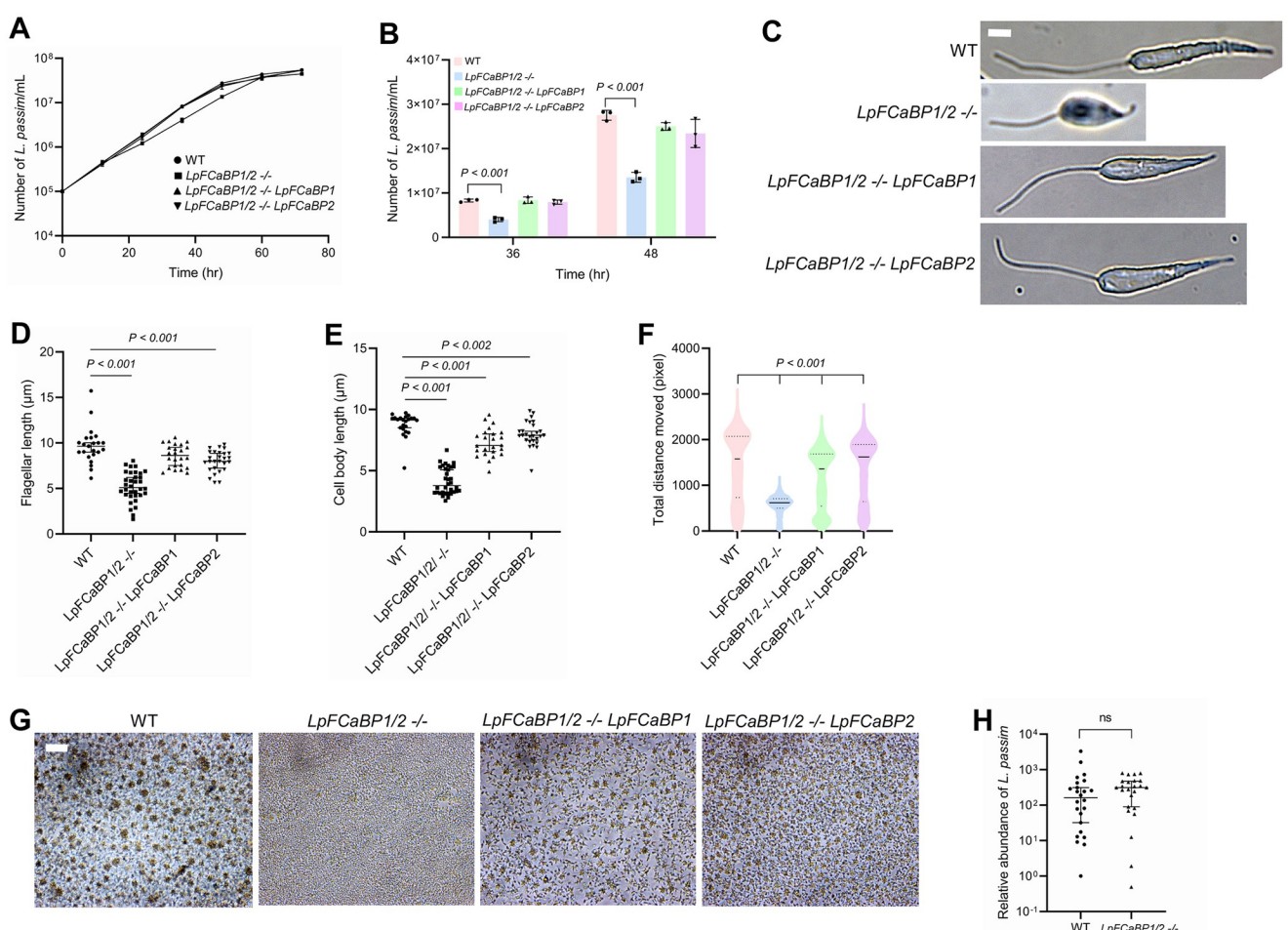

**Fig 7. Phenotypes of *LpFCaBPs* deleted mutant under culture conditions and in honey bees.** Growth rates of wild-type (WT, circle), *LpFCaBPs* homozygous mutant (clone D1, *LpFCaBP1/2 -/-*, square), *LpFCaBPs* homozygous mutant expressing either *LpFCaBP1* (*LpFCaBP1/2 -/- LpFCaBP1*, triangle) or *LpFCaBP2* (*LpFCaBP1/2 -/- LpFCaBP2*, reverse triangle) *L. passim* in the modified FP-FB medium were monitored at 29°C for three days (biological replicates, n = 3). (B) Density of WT, *LpFCaBP1/2 -/-*, *LpFCaBP1/2 -/- LpFCaBP1*, and *LpFCaBP1/2 -/- LpFCaBP2* parasites compared at 36 and 48 hours after culture. *P < 0.001* between WT and *LpFCaBP1/2 -/-* at 36 and 48 hours. Statistical comparisons were carried out using Welch's *t*-test. (C) Morphology of WT, *LpFCaBP1/2 -/-*, *LpFCaBP1/2 -/- LpFCaBP1*, and *LpFCaBP1/2 -/- LpFCaBP2* parasites. Scale bar: 2 μm. (D) Flagellar length of individual WT (n = 23), *LpFCaBP1/2 -/-* (n = 34), *LpFCaBP1/2 -/- LpFCaBP1* (n = 24), and *LpFCaBP1/2 -/- LpFCaBP2* (n = 27) parasites. *P < 0.001* between WT and *LpFCaBP1/2 -/-* or *LpFCaBP1/2 -/- LpFCaBP2*. Statistical comparison was carried out using the Steel test. (E) Cell body length of individual WT (n = 23), *LpFCaBP1/2 -/-* (n = 34), *LpFCaBP1/2 -/- LpFCaBP1* (n = 24), and *LpFCaBP1/2 -/- LpFCaBP2* (n = 27) parasites. *P < 0.001* between WT and *LpFCaBP1/2 -/-* or *LpFCaBP1/2 -/- LpFCaBP1*. *P < 0.002* between WT and *LpFCaBP1/2 -/- LpFCaBP2*. (F) Motility (total distance moved for 1 minute) of individual WT (n = 841), *LpFCaBP1/2 -/-* (n = 1777), *LpFCaBP1/2 -/- LpFCaBP1* (n = 1085), and *LpFCaBP1/2 -/- LpFCaBP2* (n = 901) parasites shown by a violin plot. Median, as well as the first and third quartiles, are indicated by solid and dashed lines, respectively. *P < 0.001* between WT and *LpFCaBP1/2 -/-*, *LpFCaBP1/2 -/- LpFCaBP1*, or *LpFCaBP1/2 -/- LpFCaBP2*. Statistical comparison was carried out using the Steel test. (G) Images of WT, *LpFCaBP1/2 -/-*, *LpFCaBP1/2 -/- LpFCaBP1*, and *LpFCaBP1/2 -/- LpFCaBP2* parasites at 96 hours after culture. The clusters of parasites represent rosettes. Scale bar: 20 μm (H)The relative abundance of *L. passim* in individual honey bees (n = 24) at 14 days after the infection compared between WT and *LpFCaBP1/2 -/-* parasites. One sample infected by the WT parasite was set as 1, and the median with 95% CI is shown. The Brunner-Munzel test was used for statistical analysis. ns: not significant.

(Fig 3). This suggests that gene duplication occurred in the common ancestor of *Trypanosoma* and Leishmaniinae. While *T. brucei*, *T. theileri*, and *L. pyrrhocoris* have further duplicated *FCaBP* genes, it appears that four *Leishmania* species and *L. seymouri* have lost *FCaBP2* by pseudogenization and deletion, respectively. When we expressed GFP fused with the sorting sequence of TbFCaBP in *L. passim*, we observed the cell body membrane localization.

Meanwhile, GFPs fused with the sorting sequences of TcFCaBP, LdFCaBP, and CfFCaBP1 were present in both the flagellum and cell body membrane (Fig 4C). These findings indicate that the BBSome complex in *L. passim* may not recognize the flagellar sorting sequence of TbFCaBP, but partially recognizes those of TcFCaBP, LdFCaBP, and CfFCaBP1. Therefore, it appears that dually acylated FCaBP is naturally targeted to the plasma membrane of the entire body, and specific N-terminal sorting sequences and the BBSome complex are required for flagellar enrichment. Additionally, the inefficient localization of GFP at the distal part of the flagellum using the TcFCaBP and LdFCaBP sorting sequences suggests a functional or structural separation in the flagellum (See also Fig 5A). Meanwhile, GFP fusion proteins may also exist in the flagellar pocket and pocket neck membranes in these parasites. The lack of functional FCaBP in *L. major* demonstrates that it is not essential for completing the life cycle in sand fly and mammalian hosts, suggesting the presence of other proteins that complement the absence of FCaBP in *L. major*.

In trypanosomatid parasites with two *FCaBP* genes, mutations have accumulated in the 5' end of the ORF, modifying the N-terminal amino acid sequences after gene duplication. As a result, FCaBP1 and FCaBP2 have different subcellular localizations at the flagellum and entire body membrane, respectively. Thus, this can be considered as one example of PSR to retain duplicated genes. However, this process may not have occurred in the five *Leishmania* species and *L. seymouri*. Without an outgroup paralogous gene or a species with a single *FCaBP* gene, it is difficult to conclude whether this represents neolocalization or sublocalization. However, sublocalization is unlikely due to the absence of *FCaBP2* in the aforementioned species. Since neolocalization to the flagellum requires co-evolution with the BBSome complex, the chance of this occurring is quite low. Therefore, we like to propose that neolocalization to the entire body membrane has occurred, and FCaBP2 in the plasma membrane of the cell body must provide some advantages for *C. fasciculata*, *L. passim*, *C. bombi*, *C. expoeki*, and *L. pyrrhocoris*.

We showed that S9 and S10 conserved among FCaBP1s across four trypanosomatid species are not only important for the efficient flagellar localization but also myristoylation and palmitoylation at G2 and C3. Similarly, T9 and Q10 of FCaBP2 appear to be essential for the acylation at the same amino acid residues (Fig 4B). Thus, myristoylation and palmitoylation at G2 and C3 in FCaBPs are likely to be context dependent influenced by the surrounding amino acid residues.

## Interaction between LpFCaBP1 and the BBSome complex in *L. passim*

The flagellar localization of LpFCaBP1 may depend on the interaction between the N-terminal sorting sequence and the BBSome complex, rather than TULP (Fig 5). These results are consistent with the requirement of the BBSome complex for the exit of dually acylated phospholipase D to the cilia in *Chlamydomonas* [40]. Although we did not detect interaction between LpFCaBP2N16-UltraID and Flag-LpBBS1, this could be due to the mislocalization of LpFCaBP2N16-UltraID at the anterior end of *L. passim*. It remains to be determined whether the entry/exit of LpFCaBP1 to the flagellum requires the BBSome and IFT complexes.

## Roles of FCaBPs in flagellar morphogenesis and motility of *L. passim*

We found that LpFCaBPs are not essential for the viability of *L. passim*. However, the loss of LpFCaBPs altered the growth characteristics of *L. passim* in the culture medium and impaired flagellar morphogenesis and motility. The formation of rosettes, which requires the clustering of parasites through flagellar movement [41], was also defective in *LpFCaBPs*-deleted parasites (Fig 7). In contrast, knocking down *TbFCaBPs* (*Calflagin Tb17*, *Tb24*, and *Tb44*) in *T. brucei* did not affect parasite growth and motility [20]. The effects of *FCaBP* loss-of-function may

differ between monoxenous parasites like *L. passim* (infecting only honey bees) and dixenous parasites like *T. brucei* (infecting both tsetse flies and mammals). It is also possible that the small amount of TbFCaBPs remaining after gene knockdown [20] is sufficient to support the normal growth and motility of *T. brucei* in the culture medium. We also found that neither FCaBP1 nor FCaBP2 alone can fully rescue the phenotypes of *LpFCaBPs*-deleted parasites (Fig 7). Assuming that the amount of ectopic LpFCaBP1 or LpFCaBP2 protein is comparable to that of the endogenous protein, *L. passim* relies on both proteins with flagellar and entire body localizations. This is consistent with a PSR hypothesis [8,12]. FCaBP likely functions as a calcium signaling protein, and it will be important to uncover how downstream effectors are involved in the growth and flagellar formation of *L. passim*.

### Roles of the flagellum in the host infection of *L. passim*

We observed that *LpFCaBPs*-deleted parasites can infect honey bee as effectively as wild-type parasites in our assay (Fig 7H). Although the mutant parasites are less active, they can reach the honey bee hindgut through normal gut flow. *L. passim* appears to attach to the hindgut wall through the structurally modified flagellum [42], and the short flagella of the mutant parasites would be sufficient for attachment. In contrast, TbFCaBPs-depleted *T. brucei* showed attenuated parasitemia in infected mice [20], but the underlying mechanism and the infection in the tsetse fly gut were not clarified. The precise *in vivo* roles of FCaBP in infecting insects and mammalian hosts remain to be answered.

## Materials and methods

### Establishing *L. passim* to express the GFP fusion proteins

To express the GFP fusion proteins LpFCaBP1-GFP, LpFCaBP2-GFP, and LpPFRP5-GFP, we amplified the complete ORFs of *LpFCaBP1*, *LpFCaBP2*, and *LpPFRP5* genes using PCR with KOD-FX DNA polymerase (TOYOBO), *L. passim* genomic DNA, and the primer pairs: LpFCaBP1-5 and LpFCaBP1-3, LpFCaBP2-5 and LpFCaBP2-3, and LpPFRP5-5 and LpPFRP5-3. The PCR products were digested with XbaI, purified from the gel, and then cloned into the XbaI site of pTrex-n-eGFP plasmid DNA [43]. To construct plasmids expressing LpFCaBP1 and LpFCaBP2 chimeric GFP fusion proteins, we amplified DNA fragments encoding the N-terminal 16 or 28 amino acids of LpFCaBP1 and LpFCaBP2 using 1444F primer and LpFCaBP1-N16-Rev, LpFCaBP1-N28-Rev, LpFCaBP2-N16-Rev, or LpFCaBP2-N28-Rev primer. Similarly, we amplified DNA fragments encoding amino acids 17–154 or 29–154 of LpFCaBP1 and LpFCaBP2 using LpFCaBP-487R primer and LpFCaBP1-N17-For, LpFCaBP1-N29-For, LpFCaBP2-N17-For, or LpFCaBP2-N29-For primer. To make the four chimeras, two DNA fragments for the corresponding N-terminal and internal amino acids of LpFCaBP1 and 2 or vice versa were ligated by fusion PCR using 1444F and LpFCaBP-487R primers. The resulting fusion PCR products and plasmids expressing LpFCaBP1-GFP or LpFCaBP2-GFP were digested with BamHI and ligated. Plasmid DNAs expressing GFPs with N-terminal sorting sequences of LpFCaBP1, LpFCaBP1K11I, LpFCaBP1TQ, LpFCaBP2, LpFCaBP2I11K, LpFCaBP2SS, TcFCaBP, TbFCaBP, LdFCaBP1, CfFCaBP1, CfFCaBP2, or LsFCaBP1 were constructed by cloning the corresponding complementary oligonucleotides into the XbaI site of pTrex-n-eGFP.

We collected actively growing *L. passim* ($4 \times 10^7$), washed twice with 5 mL PBS, and then resuspended in 0.4 mL of Cytomix buffer without EDTA (20 mM KCl, 0.15 mM CaCl$_2$, 10 mM K$_2$HPO$_4$, 25 mM HEPES and 5 mM MgCl$_2$, pH 7.6) [44,45]. The parasites were electroporated twice (1 min interval) with 10 μg of each plasmid DNA constructed above using a Gene Pulser X cell electroporator (Bio-Rad) and cuvette (2 mm gap). We set the voltage, capacitance,

and resistance at 1.5 kV, 25 μF, and infinity, respectively. The electroporated parasites were cultured in 4 mL of modified FP-FB medium [46], and then G418 (200 μg/mL, Sigma-Aldrich) was added after 24 h to select the G418-resistant clones. We washed live *L. passim* expressing the GFP fusion protein three times with 1 mL PBS, and then placed them on poly-L-lysine-coated slide glass for imaging using a NIKON eclipse Ni-U fluorescence microscope. The exposure time remained constant at 200 msec. Using Image-J, we measured the mean fluorescence of GFP in both the entire flagellum and cell body of individual parasites, as well as the background. Then, we calculated the ratio between the fluorescence in the flagellum and the cell body after subtracting the background fluorescence. Statistical comparison was performed using the Steel test.

## Analysis of microsynteny with FCaBP genes in trypanosomatid parasites

We used TBLASTN and LpFCaBP1 as a query to identify the genomic contigs containing *FCaBP* in trypanosomatid parasites. The order and orientation of genes in the microsynteny with *FCaBP* of five *Leishmania* species, *L. seymouri*, and *L. pyrrhocoris* were determined based on annotated gene lists. For *L. passim*, *C. fasciculata*, *C. bombi*, and *C. expoeki*, we performed the analysis using TBLASTN. The amino acid sequences of 11 FCaBPs (S1 Data) were aligned using Clustal Omega (https://www.ebi.ac.uk/Tools/msa/clustalo/).

## Assessing the interaction between the N-terminal sorting sequence of LpFCaBP1 or LpFCaBP2 and either LpBBS1 or LpTULP

We constructed plasmid DNA for expressing ultraID with the N-terminal sorting sequence of either LpFCaBP1 or LpFCaBP2. The DNA fragment encoding the sorting sequence was PCR amplified using 1444F and Ultra-Rev-LpFCaBP1 or 2 primers. We also PCR amplified the DNA fragment encoding ultraID using Ultra-For-LpFCaBP1 or 2 and Ultra-3-XhoI primers and used pSF3-ultraID [38] (ADDGENE: #172878) as a template. The two DNA fragments were ligated through fusion PCR followed by BamHI and XhoI digestion and cloned into the pTrex-n-eGFP plasmid at the same restriction enzyme sites. We first constructed plasmid DNA carrying triple Flag tags by inserting the annealed complementary oligonucleotides (3Flag-5 and 3Flag-3) into the XbaI and HindIII sites of tdTomato/pTREX-b [47] (ADDGENE: #68709). Then, we PCR amplified the entire ORFs of LpBBS1 and LpTULP (S2 Data) using LpBBS1-5-HindIII and LpBBS1-3-ClaI stop primers as well as LpTULP-5-HindIII and LpTULP-3-ClaI stop primers. After digesting with HindIII and ClaI, the PCR products were cloned in the same sites of the above plasmid DNA with Flag tags.

   *L. passim* was electroporated as described above with 10 μg each of LpFCaBP1N16-UltraID or LpFCaBP2N16-UltraID and either Flag-LpBBS1 or Flag-LpTULP. The parasites expressing both proteins were selected using G418 and blasticidin (50 μg/mL, Macklin). We detected the expressed proteins by immunofluorescence. The parasites were washed and mounted on a poly-L-lysine-coated 8-well chamber slide as above, fixed with 4% paraformaldehyde, permeabilized with PBS containing 0.1% TX-100 (PT), and blocked with PT containing 5% normal goat serum (PTG). The samples were incubated overnight at 4°C with rabbit anti-Myc (500-fold dilution for the ultraID fusion proteins) or rabbit anti-Flag epitope (500-fold dilution for the Flag-tagged proteins) polyclonal antibodies (Proteintech) in PTG. The samples were washed five times with PT (5 minutes for each wash) and then incubated with Alexa Fluor 488 (for the ultraID fusion proteins) or Alexa Fluor 555 (for the Flag-tagged proteins) anti-rabbit IgG antibody (ThermoFisher) for 2 hours at room temperature. After another round of washing, the samples were observed under a microscope.

We incubated the parasites ($4 \times 10^8$) in 20 mL culture medium with 50 μM biotin for 3 hours at 28˚C, followed by washing with 5 mL PBS twice. The samples were then suspended in 2 mL lysis buffer (10 mM Tris-HCl, pH 7.5, 150 mM NaCl, 0.5% NP-40, 0.5 mM EDTA) containing a protease inhibitor cocktail (Beyotime) and sonicated three times (5 seconds each) at amplitude 10 using a Q700 sonicator (Qsonica). After centrifugation, the supernatants were collected, and 50 μL of streptavidin agarose (Yeasen) and 12.5 μL of DYKDDDDK (Flag) Fab-Trap Agarose (Chromotek) were added to the half volume of each sample, respectively. The beads were washed five times (5 minutes each) with washing buffer (10 mM Tris-HCl, pH 7.5, 150 mM NaCl, 0.05% NP-40, 0.5 mM EDTA) and then suspended in 60 μL of SDS-PAGE sample buffer (2% SDS, 10% glycerol, 10% β-mercaptoethanol, 0.25% bromophenol blue, 50 mM Tris-HCl, pH 6.8). The samples were heated at 95˚C for 5 minutes, and then 7.5 μL and 30 μL of each sample precipitated by Flag Fab beads and streptoavidin beads, respectively was applied to a 10% SDS-PAGE gel. The proteins were transferred to a nitrocellulose membrane (Pall Life Sciences) and the membrane was blocked with PBST (PBS with 0.1% Tween-20) containing 5% BSA at room temperature for 30 minutes. The membrane was incubated with a 1000-fold diluted anti-Flag epitope polyclonal antibody overnight at 4˚C. After washing five times with PBST (5 minutes each), the membrane was incubated with a 10,000-fold diluted IRDye 680RD donkey anti-rabbit IgG (H+L) secondary antibody (LI-COR Biosciences) in PBST containing 5% skim milk at room temperature for 2 hours. Following another round of washing, the membrane was visualized using ChemiDoc MP (BioRad).

## Deletion of *LpFCaBP* genes by CRISPR

To delete *LpFCaBP1* and *2* genes, we designed the gRNA sequence using a custom gRNA design tool (http://grna.ctegd.uga.edu) [48]. Complementary oligonucleotides (0.1 nmole each) corresponding to the sgRNA sequences (LpFCaBP-For and LpFCaBP-Rev) were phosphorylated by T4 polynucleotide kinase (TAKARA), annealed, and cloned into BbsI-digested pSPneogRNAH vector [49]. We electroporated *L. passim* expressing Cas9 [39] with 10 μg of the constructed plasmid DNA and selected the transformants by blasticidin and G418 to establish parasites expressing both Cas9 and *LpFCaBP* gRNA.

For the construction of donor DNA for *LpFCaBP* genes, we performed fusion PCR of three DNA fragments: 5'UTR of *LpFCaBP1* (574 bp, LpFCaBP1 5'UTR-F and LpFCaBP1 5'UTR-R), the ORF of the *Hygromycin B phosphotransferase (Hph)* gene derived from pCsV1300 [50] (1026 bp, LpFCaBP1 Hph-F and LpFCaBP2 Hph-R), and 3'UTR of *FCaBP2* (590 bp, LpFCaBP2 3'UTR-F and LpFCaBP2 3'UTR-R). The fusion PCR products were cloned into the EcoRV site of pBluescript II SK(+) and the linearized plasmid DNA (10 μg) by HindIII was used for electroporation of *L. passim* expressing both Cas9 and *LpFCaBP* gRNA as described above.

After electroporation, *L. passim* resistant to blasticidin, G418, and hygromycin (150 μg/mL, Sigma-Aldrich) were selected, and single parasites were cloned by serial dilutions in a 96-well plate. The genotype of each clone was initially determined through the detection of 5' wild-type (WT) and knock-out (KO) alleles for *LpFCaBPs* by PCR. After identifying heterozygous (+/-) and homozygous (-/-) KO clones, and their 5'WT (LpFCaBP1 5'UTR-Outer-F and LpFCaBP-152R), 5'KO (LpFCaBP1 5'UTR-Outer-F and Hyg-159R), 3'WT (LpFCaBP-538F and LpFCaBP2 3'UTR-Outer-R), and 3'KO (Hyg-846F and LpFCaBP2 3'UTR-Outer-R) alleles were confirmed by PCR using the specific primer sets.

## RT-PCR

Total RNA was extracted from wild-type, *LpFCaBPs* heterozygous, and homozygous mutant parasites using TRIzol reagent (Sigma-Aldrich). Reverse transcription of 0.2 μg of total RNA

was performed using ReverTra Ace (TOYOBO) and random primers, followed by PCR with KOD-FX DNA polymerase. To specifically detect *LpFCaBP1* and *2* mRNAs, we designed reverse primers corresponding to the N-terminal sorting sequences (LpFCaBP1-36R and LpFCaBP2-36R). *L. passim* splice leader sequence (LpSL-F) was used as the forward primer. To detect *LpGAPDH* mRNA, we used LpGAPDH-F and LpGAPDH-R primers.

## Culture, flagellar and cell body length measurement, and motility measurement of *L. passim*

To construct plasmid DNA expressing *LpFCaBP1* or *LpFCaBP2*, we first PCR amplified the ORF using LpFCaBP1-5 and LpFCaBP1-3-XhoI or LpFCaBP2-5 and LpFCaBP2-3-XhoI primers. The plasmid DNA to express the corresponding GFP fusion protein was used as a template. We digested the resulting PCR products with XbaI and XhoI followed by subcloning into the same restriction enzyme sites of pTrex-n-eGFP plasmid DNA wherein neomycin resistance gene was replaced by bleomycin resistance gene. We electroporated *LpFCaBPs* homozygous mutant (*LpFCaBP1/2 -/-*) parasites with above plasmid DNA and the parasites expressing *LpFCaBP1* or *LpFCaBP2* were selected using hygromycin and zeocin (50 μg/mL, InvivoGen).

For culture, flagellar and cell body length measurement, and motility measurement, we inoculated wild-type, *LpFCaBP1/2 -/-*, and *LpFCaBP1/2 -/-* expressing either *LpFCaBP1* or *LpFCaBP2* parasites in the culture medium at $10^5$/mL at 29°C. The number of parasites was counted every 12 hours using a hemocytometer, and images of the cultured parasites were captured simultaneously for a week. We measured the length of both the flagellum and cell body of individual parasites at 60 hours after culture, utilizing phase images with Image-J. To record the movement of parasites, videos were created by taking images for 1 minute every second. The movement of parasites was tracked using TrackMate v7.10.2 [51] as a Fiji [52] plugin. The videos were imported to Fiji, converted to 8-bit grayscale, and the brightness and contrast were adjusted for better tracking. The Laplacian of Gaussian detector was used with an estimated object diameter of 30.0–36.0 pixels and a quality threshold of 0.2–0.5 for the detection of individual parasites. We used the Simple Linear Assignment Problem tracker to track the parasites by adjusting the linking maximum distance and gap-closing maximum distance to 35.0–200.0 pixels and gap-closing maximum frame gap to 1. We characterized above phenotypes with two independent clones of *LpFCaBP1/2 -/-* parasites (D1 and F6) and their phenotypes were the same. We thus used the clone D1 for further experiments.

## Honey bee infection

To infect honey bees with *L. passim*, parasites collected during the logarithmic growth phase ($5 \times 10^5$/mL) were washed with PBS and suspended in sterile 10% sucrose/PBS at $5 \times 10^4$/μL. Newly emerged honey bee workers were collected by placing the frames with late pupae in 33°C incubator and starved for 2–3 hours. Twenty individual honey bees were fed with 2 μL of the sucrose/PBS solution containing either wild-type or *LpFCaBPs* homozygous mutant ($10^5$ parasites in total). The infected honey bees were maintained in metal cages at 33°C for 14 days and then frozen at -80°C. This experiment was repeated three times. We sampled eight honey bees from each of the above three experiments, and thus analyzed 24 honey bees in total infected with either wild-type or *LpFCaBPs* homozygous mutant. Genomic DNAs were extracted from the whole abdomens of individual bees using DNAzol reagent (Thermo Fisher). We quantified *L. passim* in the infected honey bee by qPCR using LpITS2-F and LpITS2-R primers which correspond to the part of internal transcript spacer region 2 (ITS2) in the *ribosomal RNA* gene. Honey bee *AmHsTRPA* was used as the internal reference using

AmHsTRPA-F and AmHsTRPA-R primers [35]. The relative abundances of *L. passim* in the individual honey bees (24 each infected by wild-type or mutant *L. passim*) were calculated by the $\Delta C_t$ method and we set one sample infected by wild-type as 1. The statistical analysis was performed using the Brunner-Munzel test. All of the above primers are listed in S3 Data.

## Supporting information

**S1 Data. DNA and amino acid sequences of 18 FCaBPs as well as the aligned sequences.**
(DOCX)

**S2 Data. DNA and amino acid sequences of three *Trypanosoma theileri* FCaBPs, LpPFRP5, LpBBS1, and LpTULP.**
(DOCX)

**S3 Data. List of primers used in this study.**
(DOCX)

**S4 Data. Ratios of mean GFP fluorescence of flagellum to that of cell body.**
(XLSX)

**S5 Data. Numerical data shown in Fig 7.**
(XLSX)

**S1 Video. Movement of wild-type *L. passim*.**
(MP4)

**S2 Video. Movement of LpFCaBPs deleted mutant *L. passim*.**
(MP4)

**S3 Video. Movement of LpFCaBPs deleted mutant *L. passim* expressing LpFCaBP1.**
(MP4)

**S4 Video. Movement of LpFCaBPs deleted mutant *L. passim* expressing LpFCaBP2.**
(MP4)

## Acknowledgments

We thank Yizhen Shao for his contribution to this study.

## Author Contributions

**Conceptualization:** Tatsuhiko Kadowaki.

**Formal analysis:** Xuye Yuan, Tatsuhiko Kadowaki.

**Funding acquisition:** Tatsuhiko Kadowaki.

**Investigation:** Xuye Yuan, Tatsuhiko Kadowaki.

**Supervision:** Tatsuhiko Kadowaki.

**Writing – original draft:** Tatsuhiko Kadowaki.

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
