## [Decision Letter · Decision Letter 0]

12 Oct 2023

Dear Dr Kadowaki,

Thank you very much for submitting your Research Article entitled 'Protein subcellular relocalization and function of duplicated flagellar calcium binding protein genes in honey bee trypanosomatid parasite' to PLOS Genetics.

The manuscript was fully evaluated at the editorial level and by independent peer reviewers. The reviewers appreciated the attention to an important and interesting problem, but raised some substantial concerns about the current manuscript. For example, while Reviewer 1 agreed that the data presented does support that LPFCaBP1 and 2 are differentially localized via PSR, there were concerns that the situation is more complex and that more functional data on the paralogs needs to be provided.  Reviewer 2 acknowledged the high quality of the cell data but had concerns regarding the depth of the evolutionary analysis and discussion. Based on the reviews, we will not be able to accept this version of the manuscript, but we would be willing to review a much-revised version. We cannot, of course, promise publication at that time.

If you decide to revise the manuscript for further consideration at PLOS Genetics, please aim to resubmit within the next 60 days, unless it will take extra time to address the concerns of the reviewers, in which case we would appreciate an expected resubmission date by email to plosgenetics@plos.org.

Please do not hesitate to contact us if you have any concerns or questions.

Yours sincerely,

Ashley Soyong Byun, PhD

Guest Editor

PLOS Genetics

Eva Stukenbrock

Section Editor

PLOS Genetics

Reviewer's Responses to Questions

**Comments to the Authors:**

Reviewer #1: Overview

This study seeks to characterises the location, localisation-determinants and function of two tandemly-duplicated genes in the trypanosomatid parasite of bees, Lotmaria passim, with the aim of understanding more about the processes driving/maintaining gene duplication and diversification. These proteins are highly similar, with the main differences occurring towards the N terminus, close to the myristate and palmitate addition signals. The authors express tagged proteins, and protein chimeras to demonstrate that the paralogs have different localisation distributions and that this is primarily conferred by the N terminal sequences. By expressing the N terminal peptide of the flagellar-localised paralog fused to an enzyme that performs promiscuous biotinylation followed by streptavidin purification of biotinylated proteins and western blot analysis, the authors present evidence that flagellar localisation of LpFCaBP1 is done via the N-terminal 16 residues interacting with interacting with BBS1. The authors perform KO experiments to show that although an effect is observed in vitro, KO parasites can still infect bees, suggesting a non-essential role in vivo.

Although the claim that the differential distributions of the two proteins is (predominantly) conferred by the N terminal sequences is solid, differential localisation of paralogs is not new and several of the descriptions and conclusions in the text do not tie with the data as shown in the Figures. Most importantly, however, the authors give little functional insights into these proteins, and the functional significance, if any, of their different localisation distributions is not investigated.

Main issues:

Line 126: “The N-terminal 16 amino acids of LpFCaBP1 and LpFCaBP2 were sufficient to direct GFP to the flagellum and cell body, respectively (Fig. 2)”.

Although this statement is justified by the experiment fusing the LmFCaBP1/2(1-16) to GFP, the other data appears to be more complex than the authors suggest in the text. For example, while the LpFCaBP1N28/FCaBP2-GFP does indeed appear to reproduce the LpFCaBP1-GFP full length protein, the LpFCaBP1N16/FcaBP2 does not, as this latter chimera appears to be missing from the cell-body membrane altogether. Would this not suggest that there is an additional signal in LpFCaBP1(N16-28) that facilitates some distribution to the cell-body membrane?

Additionally, both of the other chimeras (LpFCaBP2N28/FCaBP1-GFP and LpFCaBP2N16/FCaBP1-GFP) are missing from the flagellar membrane (at least in the example given), in contrast to FCaBP2-GFP. Would this not suggest that there are sequences in FCaBP2(16-204) that confer at least some flagellar membrane localisation?

Line 152 – 170

the authors express the N terminal region of a series of calmodulin homologs from different trypanosomatid species and note different localisation distributions. In particular, they note that expressing the N24 residues of Tb/TcFCaBP localises to the cell body membrane, and not the flagellar membrane as has been shown for the full length protein in T. brucei/cruzi. From this they conclude that a the default localisation is the cell body membrane, and that a trans-acting factor causes localisation to the flagellar membrane. Beyond showing that the N terminal peptide clearly causes differences in localisation distribution (already known from Figure 2) I am unconvinced by this line of experiments and the conclusions that were drawn because of the biological irrelevance of drawing conclusions when expressing orthologs from diverse species and lack of controls (e.g. expression in native species, expression of full length protein). A more useful line of enquiry would have been to use these “natural mutation experiment” as a basis to perform directed mutagenesis to obtain mechanistic insight into the different sorting signals.

Figure 5 and line 171-186.

The authors show that BBS1 appears to be somewhat biotinylated by LpFCaBP1N16-UltraID. However, the authors do not show that the an interaction of LpFCaBP1N16 (or, indeed, LpFCaBP1N16) with the BBSome is necessary for its flagellar localisation, for example with BBS KO cells. We should also note that this experiment was only done with the N terminal peptide, which is less biologically realistic than the full length protein

Although using Flag-LpTULP adds an important control, the BBS1 band is not very convincing and this data would be strengthened using label-free using mass spectrometry to confirm BBS1 presence and relative abundance.

Note that biotinylation does not necessarily indicate “interaction” but could simply indicate proximity.

I would advise the authors to investigate any interaction of the BBSome with LpFCaBP2, because the BBSome has been implicated in removal of proteins from the flagellum

Figure 7 and line 187-214:

The authors investigate the functional characterises of KO parasites. Here I think the authors have missed a trick. Given the emphasis on understanding paralogs with different localisations, performing a addback with the different paralogs individually would seem to be a good way of getting at their potential different function.

Additionally, further analysis of the KO parasites seem warranted, for example performing TEM on flagellar cross sections to detect defects, and a microscopical analysis of the KO parasites from in vivo infections, as a PCR-based assay seems to be rather coarse and may have missed a phenotype.

Minor points:

Line 115: “These findings indicate that the two duplicated FCaBP genes encode proteins with distinct subcellular localizations”.

The localisations are actually the same (flagellar membrane and cell-body membrane), but the key factor is their relative enrichment on each plasma-membrane sub-domain. Suggest to re-phrase as “differentially enriched, but overlapping, distributions”, or something that conveys the same idea

The miscopy figures all need scale bars. In several instances the cells appear to be different magnifications – would recommend standardising the magnification within a Figure.

Unclear in some cases if it DOES localise a cell-body/flagellar membrane subdomain, but is difficult to discern because of the high levels of expression of the other subdomain (e.g. Figure 1 bottom two rows, but also perhaps in other places where there is strong expression to a particular subdomain). Can deal with this in a multiplicity of ways, but some quantification would be helpful

Figure 1:

FCaBP2-GFP needs a better example exhibiting a clear edge effect to show it is on the flagellar membrane

Provide clearer annotation for the unfamiliar reader to point out the flagellar membrane and cell body membrane

Figure 6

Can the authors clarify whether the screening primers are specific to each paralog?

Figure 7A and B:

Clarify on whether the growth curve data and statistics are based on technical or biological replicates. If technical replicates (i.e. the same experiment done in triplicate) then the very mild phenotype observed could be simply due to differences in the starting “health” of the cultures, and not reproducible.

Clarify how the length of the flagellum was calculated (e.g. measured from phase images?)

Reviewer #2: In the trypanosomatid parasites gene duplications are common and likely represent the evolution of proteins with novel functions. In this manuscript the authors investigate the duplication of a calcium binding protein in Lotmaria passim. They demonstrate that one protein localises to the flagellum and the other to the cell body and that one or the other are required for flagellum assembly and function. Overall, the cell biology data presented is of a high quality and generally supports the conclusions the authors make; however, the evolutionary analysis focussed on a restricted range of organisms, with limited discussion.

Abstract ln 30/31 – the authors do not present any data to show that the Tbrucei and Ldonovani FCaBPs do not interact with the BBSome. These leader sequences do not direct GFP to the Lpassim flagellum but that might not be due to the lack of BBSome interaction. Therefore, it is a stretch to conclude that this is due specific co-evolution between the Lotmaria BBSome and FCaBP. To really conclude this, they would need to demonstrate interaction of paired FCaBPs with BBSomes i.e. does Tbrucei FCaBP interact with Tbrucei BBSome in Tbrucei.

This point leads onto my concern with the genome/sequence analysis in this paper, which could be improved. The authors look at the conservation/synteny etc. in a range of organisms but these are all relatively closely related – are FCaBP found in more distantly related organisms such as Paratrypanosoma or Bodo saltans? If so, does that help refine the relationship between the different species?

Have the authors examined the phylogenetic relationships between these genes? Do they match the speciation of these organisms? Can this help to inform the analysis? The authors rely on the reference genomes for these organisms as presented but are they sure that the loss of FCaBP2 in Lseymouri is real and not an issue with genome assembly/annotation – is there a pseudogene downstream of FCaBP1 in this organism? In Lmexicana etc FCaBP2 is described as a pseudogene but the genome describes this as a normal protein-coding gene just without the N-terminus found in other species and the transcriptomic evidence suggests it is expressed. This protein may be perfectly functional and in fact might have a different localisation and represent a specific evolution event in these species. Where does this protein localise? In Lmajor both genes appear to pseudogenes but is this backed up by transcriptomic datasets? I would urge caution of taking what is presented in genomes at face value, especially if it looks different to what is seen in other organisms – it maybe true but might be an issue with the automated assembly of the genome.

Does the alignment of these leader sequences suggest what is important for flagellum retention/exclusion – the dipeptide SS – TQ could be potentially important? Is this a charge effect there are lots of lysines, which are supposed to be important for flagellum localisation in Tbrucei/Tcruzi but those that do not localise to the Lpassim flagellum have a number of asparagine/glutamine residues in the leader sequence – is there any pattern in the overall charge of these sequences and their localisation?

Minor comments

Need to be specific with descriptions of localisation – presumably the proteins localise to the flagellum membrane and cell body membrane, not the cell body.

The localisation of PFR5 shown here is not how a tagged PFR protein would normally look like. Is the PFR discontinuous in Lotmaria?

Can you include geneIDs in the figure legends so it is easier to find the genes they are examining.

What do the colours represent in figure 3?

Ln169/170 – the authors suggest that flagellum localisation requires sorting into the flagellum but this could equally be about retrieval of FCaBP2 from the flagellum – additional nuances should be discussed about what might define the localisation of a protein.

There was a limited information about the blots in figure 5. What was the loading of each lane, how many cell equivalents? What is the relationship between the input and streptavidin material? Without this information it is hard to judge how strong/likely this interaction is. In the discussion, there is a caveat about the weak signal detected here. Additional evidence to show this was a reciprocal interaction with FCaBP1 detected when BBSome1 was tagged with ultraID would increase confidence in this conclusion.

The functional analysis used a cell line in which both FCaBP genes had been deleted. How many clones were examined? Can you add this information to the legends etc? By deleting both genes, it is not possible to determine the relative contribution of each gene to the phenotype. Have the authors deleted each gene individually? Could they add back FCaBP1 into the deletion line and see if this restores the phenotype? Without this, it is not possible to determine if these proteins actually have separate functions as would be suggested by the PSR hypothesis.

Ln261/262 – Lmajor and Lpassim have evolved separately for millions of years and have very different life cycles etc, so I think the dispensability of FCaBP in Lpassim is likely unrelated to the ability of Lmajor to survive without these genes.

**Have all data underlying the figures and results presented in the manuscript been provided?**

Reviewer #1: Yes

Reviewer #2: Yes

PLOS authors have the option to publish the peer review history of their article (what does this mean?). If published, this will include your full peer review and any attached files.

Reviewer #1: No

Reviewer #2: No

---

## [Decision Letter · Decision Letter 1]

14 Feb 2024

Dear Dr Kadowaki,

Thank you very much for submitting your Research Article entitled 'Protein subcellular relocalization and function of duplicated flagellar calcium binding protein genes in honey bee trypanosomatid parasite' to PLOS Genetics.

The manuscript was fully evaluated at the editorial level and by independent peer reviewers. The reviewers remarked that the manuscript was greatly improved through your addition of data and more thorough analyses but identified some minor concerns that we ask you address in a revised manuscript.

We therefore ask you to modify the manuscript according to the review recommendations. Your revisions should address the specific points made by each reviewer.

Yours sincerely,

Ashley Soyong Byun, PhD

Guest Editor

PLOS Genetics

Eva Stukenbrock

Section Editor

PLOS Genetics

Reviewer's Responses to Questions

**Comments to the Authors:**

Reviewer #1: The authors have done a substantial amount of work and the manuscript is greatly improved.

Revisions:

Line 343: “We also found that either FCaBP1 or FCaBP2 alone can not fully rescue the phenotypes of LpFCaBPs-deleted parasites (Fig. 7), suggesting that L. passim depends on both proteins with the flagellar and entire body localization”

The authors should acknowledge (or rule out if they have data to indicate otherwise, such as a western blot) that the inability to fully rescue the KO phenotype could be because the individual FCaBP1 or FCaBP2 addbacks do not express as highly as the combined FCaBP1/ FCaBP2 in the wild type.

The microscopy still needs scale bars

Reviewer #2: On the whole the authors have suitably addressed my concerns. They have provided a more thorough genomic analysis and have provided additional functional data by examining LpFCaBP1/2 add back parasites to define their relative roles. However, while the inclusion of the enrichment analysis has improved the manuscript and helps to interpret their findings they need to more clearly explain the results and provide more details about the methodology.

More detail about measurement of fluorescence needed – was there a background correction? Was all the cell body/flagellum measured or just a part of it? What microscope was used? Was a consistent exposure time used etc.?

Figure 1 - If the ratio of LpFCaBP2 between flagellum and cell body is 0.92±0.28 then it is essentially evenly distributed between the flagellum and cell body and at the moment the manuscript suggests that it is enriched in the cell body.

Figure 2 - For the LpFCaBP1N16/FCaBP2-GFP fusion how was the measurement of the flagellum fluorescence intensity performed was this the entire flagellum or just the proximal domain? On a related note this localisation pattern could represent an enrichment within the flagellar pocket and pocket neck membranes which constitute another membrane domain in these parasites, rather than the proximal domain of the flagellum.

Figure 2 - The authors need to describe the localisation pattern more clearly. The enrichment analysis shows no difference but the LpFCaBP1N16/FCaBP2-GFP shows an enrichment in the proximal flagellum region whereas the LpFCaBP1N28/FCaBP2-GFP is enriched along the entire flagellum as is the LpFCaBP1N16-GFP fusion and this is not pointed out. Moreover, as the LpFCaBP2N16-GFP gives a different enrichment ratio to LpFCaBP2-GFP the authors are correct in stating that the EF hands likely have an influence on protein localisation but this does not come across well in the text. At what point in the proteins does the EF hand domain start? This should be indicated somewhere in the manuscript.

Line 146-9 – the authors should indicate that this protein would likely be in the cytosol as they have no evidence to indicate its localisation.

Figure 4 - As the SS/TQ mutants appear to be cytosolic the key conclusion which the authors make is that this seems to affect fatty acid modification but because of that the enrichment ratio is not overly informative as it is unlikely that the differential localisation is occurring through the same mechanism.

Line 190 – the conclusion here needs to be modified as the results show that K11 is not necessary for flagellum localisation of LpFCaBP1N16. It appears therefore to contribute to LpFCaBP2N16 flagellum localisation but is not necessary for LpFCaBP1N16 flagellum localisation.

Figure 4C – the authors point out the proximal flagellum enrichment of the Tc/Ld FCaBP leader sequences and again this relates to my comment about the flagellar pocket membrane domain that could be added to the discussion. The authors comment on the inefficient localisation (lines 298-300) but they should acknowledge the present of another membrane domain here as well.

Line 274 – can the authors clarify this sentence, it is unclear what they mean by function is the same as a calcium binding protein.

Finally, the language of the paper needs to be improved throughout and the manuscript should be read by a native speaker before publication.

**Have all data underlying the figures and results presented in the manuscript been provided?**

Reviewer #1: Yes

Reviewer #2: **No: **The numerical data for the graphs in figure 7 and enrichment analysis was not provided.

PLOS authors have the option to publish the peer review history of their article (what does this mean?). If published, this will include your full peer review and any attached files.

Reviewer #1: No

Reviewer #2: No

---

## [Editor Report · Decision Letter 2]

23 Feb 2024

Dear Dr Kadowaki,

We are pleased to inform you that your manuscript entitled "Protein subcellular relocalization and function of duplicated flagellar calcium binding protein genes in honey bee trypanosomatid parasite" has been editorially accepted for publication in PLOS Genetics. Congratulations!

Yours sincerely,

Ashley Soyong Byun, PhD

Guest Editor

PLOS Genetics

Eva Stukenbrock

Section Editor

PLOS Genetics

Comments from the reviewers (if applicable):

**Data Deposition**

http://datadryad.org/submit?journalID=pgenetics&manu=PGENETICS-D-23-00696R2

**Press Queries**

---

## [Editor Report · Acceptance letter]

26 Feb 2024

PGENETICS-D-23-00696R2 

Protein subcellular relocalization and function of duplicated flagellar calcium binding protein genes in honey bee trypanosomatid parasite 

Dear Dr Kadowaki, 

We are pleased to inform you that your manuscript entitled "Protein subcellular relocalization and function of duplicated flagellar calcium binding protein genes in honey bee trypanosomatid parasite" has been formally accepted for publication in PLOS Genetics! Your manuscript is now with our production department and you will be notified of the publication date in due course.

With kind regards,

Zsofia Freund

PLOS Genetics

On behalf of:
